# Modeling and Optimization of Geraniol ((2E)-3,7-Dimethyl-2,6-Octadiene-l-ol) Transformation Process Using Response Surface Methodology (RSM)

**Anna Fajdek-Bieda** , **Andrzej Perec *** and **Aleksandra Radomska-Zalas**

Faculty of Technology, Jacob of Paradies University, Teatralna 25, 66-400 Gorzow Wielkopolski, Poland
* Correspondence: aperec@ajp.edu.pl

**Abstract:** This paper presents the modeling of the geraniol transformation process using response surface methodology (RSM). It uses a combination of both statistical and mathematical modeling methods to study the relationships occurring between several explanatory variables and one or more response variables. Interactions occurring between process variables are studied using statistical techniques. In this paper, the influence of the most important process parameters, such as temperature 20–110 °C, catalyst concentration (mironecuton) 1.0–5.0 (wt.%), and reaction time 0.25–2 (h), is presented. The response functions were the conversion of geraniol (GA), the selectivity of conversion to thumbergol (TH), and the selectivity of conversion to 6,11-dimethyl-2,6,10-dodecatriene-1-ol (DMC). In addition, the effects of all control parameters on each of the response parameters were presented in the form of second-order polynomials. Attempts were made to identify process conditions that would allow high values of the process function.

**Keywords:** reaction time; thumbergol; temperature; modeling; geraniol; conversion





## 1. Introduction

To optimize the transformation process, the response surface methodology (RSM) is used, which uses methods of mathematical and statistical analysis to determine the interaction between the variables under study, allowing the determination of the correct response with a minimum number of experiments [1]. The RSM was proposed in the mid-20th century by Box and Wilson, and a full study can be found in [2,3].

In practice, the RSM is used in a sequential manner [4,5]. Conducting an experiment and then interpreting the results makes it possible to assess the influence of the various factors under study on the value of the response surface [6–8].

RSM is one of the more commonly used metamodeling methods, the purpose of which is to approximate the response of the model on the basis of selected values of input signals [1,2].

The RSM is one of the innovative methods used in the optimization process in many branches of the industry. Its universality is mainly due to the small number of tests used and its affordability. Additionally, there is usually more than one important answer, so problems must be optimized at the same time.

The RSM was used in modeling the efficiency of abrasive waterjet machining with rock materials [3] such as limestone [4], marble [5], phenolic composite [6], aluminum alloys [7,8], and heavy to machining metals [9], for example, Monel [10] or special steels [11,12]. Radomska-Zalas et al. applied this method to optimize the effect of cutting parameters on aluminum alloy and micro-alloy steel [13].

Gangil et al. [14] presented a study of parameter optimization in the EDM (electrical discharge machining) process to achieve maximum productivity and minimum surface roughness. For this purpose, they used a combination of RSM and VIKOR methods (a multi-criteria decision-making method).

The RSM was used in the substrate treatment process during the production of biogas energy in anaerobic digestion. A central composite system was used to develop the experiment. The RSM was used in the so-called intelligent modeling of substrate treatment in the biogas production process. The kinetics of the anaerobic fermentation process was investigated using five kinetic models. The obtained results show that among the three assessed process parameters, the temperature has the highest influence on the substrate pretreatment process. The AFIS (automated fingerprint identification system), ANN (artificial neural networks), and UAN (user action notation) models proved to be effective in modeling the anaerobic digestion process [15].

Manmai et al. [16] presented RSM as an effective method for predicting modeling and optimization of sugar and energy reduction methods. All experiments used a statistical design to develop a statistical model for multivariate analysis, which provides consideration of the effects of various parameters on the process and describes the optimal values of these variables to optimize the response [17].

Igwegbe et al. [18] investigated the effect of Picralima nitida extract (PNE) in the bio-coagulation–flocculation (BCF) process on the reduction of chemical oxygen demand (COD) in municipal waste leachate (MSWL). In the described process, the RSM and ANN methods were used. The analysis of variance that was carried out showed that the CMS model (coastal modeling system) was statistically relevant in the interpretation of the processes in the studied range. Both the UAN and ANN were able to predict the COD reduction process. Additionally, the ANN method was burdened with less error [19].

In a paper by Husien et al. [20], the application of the RSM was presented in the process of fluoride removal. The influence of the following variables was investigated: pH, time, temperature, and dose of La-FeO$_3$-NP and fluoride. The CCD (central composite design) plan was used in the UAN analysis. Performance was assessed by the regression coefficient (R$^2$), RMSE (root mean square error), SEP (standard error prediction), and AAD (average absolute deviation). All methods yielded very similar values for the optimization of the fluoride reduction process with La-FeO$_3$-NP [21].

The use of the RSM has been widely described in publications on the optimization of the conditions for the removal of trihalomethanes (THM) and natural organic matter (NOM) from drinking water. The Box–Behnken experimental model in conjunction with the RSM was used to predict the THM and NOM content in drinking water. Process variables were sMNP (the synthesis of magnetic nano-adsorbent) concentration (0.1 to 5 g), pH (4–10), and response time (5 to 90 min). To determine the adequacy of the developed model, a statistical analysis of variance (ANOVA) was performed. Additionally, the risk analysis showed that under the optimized conditions obtained by the RSM, a significant reduction in the risk of THM cancer was observed in both studied groups [22,23].

The applied UAN was used in the application of saturated polyester (BUP) biotherapy. After optimizing the solution to the problems to follow, an experimental method of responding to the solution led to reaching a state of 17, as a result of both NN (neutral network) and RSM [24,25].

RSM based on central rotary complex design (CCRD) was used to optimize the chemical coagulation process. Using the analysis of variance, square models of color reduction and TSS removal with coefficients of determination R$^2$ > 96 were obtained. Under optimal conditions, the efficiency of both color and total suspension (TSS) removal was 85% and 82%, respectively [26].

In contrast, Gupta et al. applied RSM for optimal parameter selection of electronic packaging [27].

In the paper of Fajdek-Bieda et al. [28], the optimization of the epoxidation of crotyl alcohol with the use of a titanium-silicate Ti-MWW catalyst was presented. The process was carried out under atmospheric pressure. The influence of the most important parameters of the process was investigated: temperature, molar ratio of crotyl alcohol to H$_2$O$_2$, methanol concentration, concentration of Ti-MWW catalyst, and reaction time. The response functions characterizing the epoxidation process were: the selectivity of conversion to 2,3-epoxybutan-

1-ol overreacting croton alcohol—S2,3EB1O/CA, conversion of crotyl alcohol, $C_{CA}$, and yield of 2,3-epoxybutan-1-ol. Based on the calculated signal-to-noise ratios for each process parameter, their influence on the output parameters was determined. Moreover, the optimal conditions for carrying out the process were determined to achieve the maximum. Additionally, an empirical verification of the process was carried out by comparing the two initial parameters predicted and obtained in the research. Summarizing, this method allows to limit the number of tests required to obtain the desired results, shorten the time needed to obtain them, and thus lower the costs.

Based on the RSM, the optimization of the geraniol transformation process was also conducted [29]. The impact of the following process control parameters: temperature, catalysts' concentration (clinoptilolite), and time, was presented. The output parameters describing the process were as follows: conversion of geraniol, selectivity of thumbergol, and selectivity of 6,11-dimethyl-2,6,10-dodecatrien-1-ol. The impact of control factors on all output factors as the second-degree polynomial equations was shown.

Since in the full factorial design and response surface methodology (FFD-RSM), the key factor levels are set completely independent of each other, this approach was also used in other studies.

Sen et al. [30], using the FFD and RSM, presented three factors at three levels to determine the effect of operational variables, such as feed rate, centrifugal force, and fluidization water flow rate, on the efficiency of the Knelson concentrator for chromite ore beneficiation. The quadratic models were developed to predict the $Cr_2O_3$ concentrate grade and recovery as the process responses. The results suggest that all the variables affect the grade and recovery of the $Cr_2O_3$ concentrate.

Rahman et al. [31] investigated the combined effects of temperature and time of a double-boiling treatment on the quality of Kelulut honey using the FFD-RSM approach. A three-level factorial design employing nine runs with duplicates under different combinations of temperature and time was developed. The quality of Kelulut honey in this experiment was analyzed based on physicochemical and nutritional properties. This study was a novel optimization of combined temperature and time of a double-boiling treatment.

Ebadi et al. [32] presented research on the optimized condition for desalination of the reverse osmosis (RO)-rejected stream from the Esfahan Oil Refining Company (EORC) using direct contact membrane distillation (DCMD), with polytetrafluoroethylene (PTFE) membrane as the subject. The authors used RSM and FFD modeling, carried out in a laboratory-scale set-up. Statistical criteria for validation, significance, accuracy, and adequacy confirmed the suitability of the employed quadratic polynomial model.

The FFD-RSM presented in this paper is innovative in the process of geraniol transformation with the use of mironecuton as a catalyst. The proposed method allows for the correct selection of optimal process parameters, such as temperature 20–110 °C, catalyst concentration (mironecuton) 1.0–5.0 (wt.%), and reaction time 0.25–2 (h), due to the obtained high values of the selectivity coefficients of the synthesis products and raw materials' conversion.

## 2. Results and Discussion

### 2.1. Impact of Control Factors on Geraniol Conversion

A detailed geraniol ANOVA was performed for a 95% confidence level ($\alpha = 0.05$) (Table 1). The model factor is significant when it achieves a *p*-value $< 0.05$. The correlation coefficient ($R^2$) and the adjusted correlation coefficient ($R^2_{adj}$) were used to determine the accuracy of the model. The $R^2$ coefficient was 0.963 and $R^2_{adj}$ was 0.943, as shown in Table 1. Hence, the model explained 94.3% of the variance of the data. In addition, differences between $R^2$ and $R^2_{adj}$ were lower than 0.2 for all the response variables, showing that the response surface accurately mirrors the data.

**Table 1.** ANOVA results for geraniol conversion.

| Source | DF | Adj SS | Adj MS | F-Value | *p*-Value | VIF |
|---|---|---|---|---|---|---|
| Model | 9 | 3046.58 | 338.51 | 48.88 | 0.000 | |
| Linear | 3 | 2237.39 | 745.80 | 107.70 | 0.000 | - |
| Temperature (°C) | 1 | 1051.48 | 1051.48 | 151.85 | 0.000 | 1.02 |
| Catalyst concentration (wt.%) | 1 | 648.80 | 648.80 | 93.69 | 0.000 | 1.03 |
| Time (h) | 1 | 555.18 | 555.18 | 80.17 | 0.000 | 1.02 |
| Square | 3 | 370.17 | 123.39 | 17.82 | 0.000 | - |
| Temperature (°C) * Temperature (°C) | 1 | 367.24 | 367.24 | 53.03 | 0.000 | 1.00 |
| Catalyst concentration (wt.%) * Catalyst concentration (wt.%) | 1 | 0.65 | 0.65 | 0.09 | 0.762 | 1.02 |
| Time (h) * Time (h) | 1 | 2.28 | 2.28 | 0.33 | 0.573 | 1.01 |
| Two-Way Interaction | 3 | 258.80 | 86.27 | 12.46 | 0.000 | - |
| Temperature (°C) * Catalyst concentration (wt.%) | 1 | 81.56 | 81.56 | 11.78 | 0.003 | 1.01 |
| Temperature (°C) * Time (h) | 1 | 114.72 | 114.72 | 16.57 | 0.001 | 1.01 |
| Catalyst concentration (wt.%) * Time (h) | 1 | 62.53 | 62.53 | 9.03 | 0.008 | 1.01 |
| Error | 17 | 117.72 | 6.92 | | | |
| Total | 26 | 3164.30 | | | | |
| S = 2.63147 | $R^2$ = 96.28% | | $R^2_{(adj)}$ = 94.31% | | $R^2_{(pred)}$ = 90.06% | |

The impact and correlation of each independent variable to the geraniol conversion was illustrated in a Pareto chart (Figure 1), constructed from the obtained regression polynomial equation (Equation (1)).

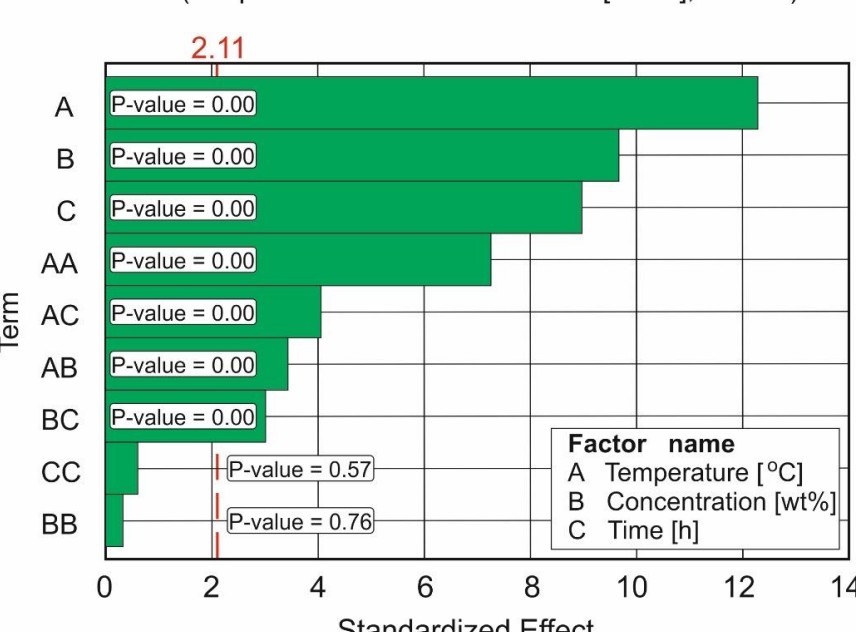

**Figure 1.** Levels of important and not important factors of the geraniol conversion for $\alpha$ = 0.05.

The standardized effect of all studied independent variables and interactions between these variables is shown in Figure 1. The standardized effect is the minimum level for showing the influence of each variable. The size of the effect corresponds to the length of the bar. Independent variables are predicted to have a statistically significant effect and play an important role in the response if the bar of the standardized effect exceeds the minimum limit, in this case 2.11, shown as a vertical red dotted line.

Regression equation in uncoded units:

$$CG = 30.07 + 0.8542\ T + 6.85\ C + 17.18\ \tau - 0.0039\ T^2 - 0.0286\ T{\cdot}C - 0.078\ T{\cdot}\tau - 1.287\ C{\cdot}\tau \qquad (1)$$

where:

CG is conversion of geraniol (wt.%),

T is temperature (°C),

C is concentration (wt.%),

τ is time (h).

To approximate the multicollinearity level, the variance inflation factor (VIF) was determined. It quantifies the multicollinearity intensity. The VIF reveals how much the variance of the assessed regression factor is inflated because of the presence of multicollinearity in the model. When the VIF is 1.0, multicollinearity is not present. No significant multicollinearity was observed for all factors tested, as the VIF was in the interval {1, 1.03}.

Figure 2a–i present the impact of individual control factors on the geraniol conversion's selectivity. The geraniol conversion takes the smallest values for the minimum catalyst concentration, lowest temperature, and shortest reaction time. The increase of the concentration and the reaction time occurred with the increase of the geraniol conversion. In this case, a directly proportional relationship was observed. Maximum geraniol conversion (97 mol%) was observed for the maximal temperature and maximal catalyst concentration, with a reaction time equal to two hours.

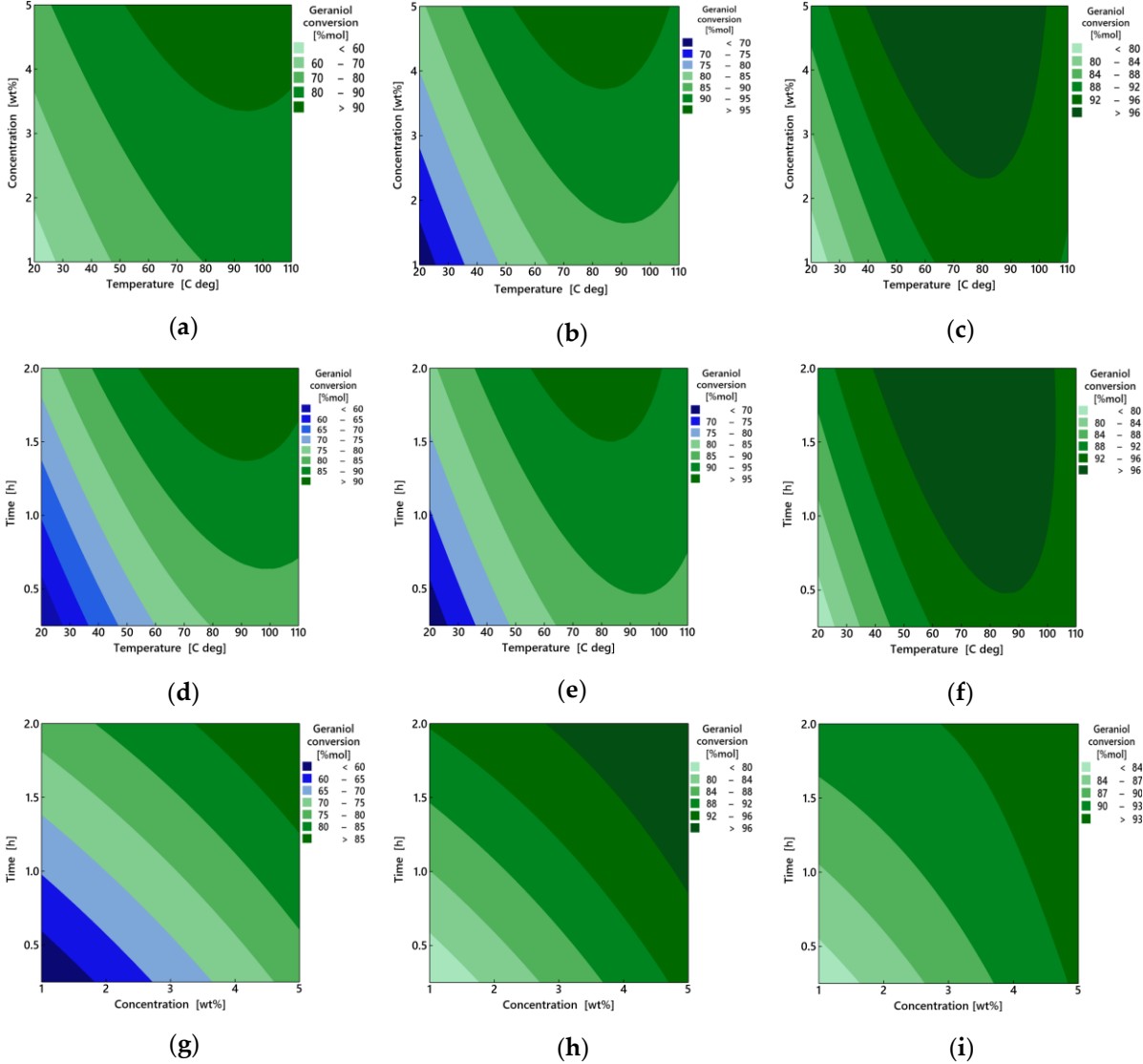

**Figure 2.** Contour plot of geraniol conversion at: time: (**a**) 0.25 h, (**b**) 1.125 h, and (**c**) 2 h, concentration: (**d**) 1 wt.%, (**e**) 3 wt.%, and (**f**) 5 wt.%, and temperature: (**g**) 20 °C, (**h**) 65 °C, and (**i**) 110 °C.

## 2.2. Impact of Control Factors on Dimethyl Selectivity

A detailed dimethyl selectivity analysis was performed by ANOVA for 95% confidence at $\alpha = 0.05$ (Table 2). The factors of the model were considered significant when their *p*-values exceeded 0.05. Here, the $R^2$ correlation coefficient and the $R^2_{adj}$ adjusted correlation coefficient were used to establish the accuracy of the model. The $R^2$ coefficient was 0.946 and $R^2_{adj}$ was 0.919, as shown in Table 2. Thus, the model explained 91.9% of the variance of the data.

**Table 2.** Analysis of variance of dimethyl selectivity.

| Source | DF | Adj SS | Adj MS | F-Value | *p*-Value | VIF |
|---|---|---|---|---|---|---|
| Model | 9 | 3684.42 | 409.38 | 32.89 | 0.000 | - |
| Linear | 3 | 3066.41 | 1022.14 | 82.13 | 0.000 | - |
| Temperature (°C) | 1 | 585.17 | 585.17 | 47.02 | 0.000 | 1.02 |
| Catalyst concentration (wt.%) | 1 | 535.48 | 535.48 | 43.02 | 0.000 | 1.03 |
| Time (h) | 1 | 1967.89 | 1967.89 | 158.12 | 0.000 | 1.02 |
| Square | 3 | 291.60 | 97.20 | 7.81 | 0.002 | - |
| Temperature (°C) * Temperature (°C) | 1 | 236.89 | 236.89 | 19.03 | 0.000 | 1.00 |
| Catalyst concentration * Catalyst concentration (wt.%) | 1 | 50.96 | 50.96 | 4.09 | 0.059 | 1.02 |
| Time (h) * Time (h) | 1 | 3.75 | 3.75 | 0.30 | 0.590 | 1.01 |
| Two-Way Interaction | 3 | 121.98 | 40.66 | 3.27 | 0.047 | - |
| Temperature (°C) * Catalyst concentration (wt.%) | 1 | 19.58 | 19.58 | 1.57 | 0.227 | 1.01 |
| Temperature (°C) * Time (h) | 1 | 5.36 | 5.36 | 0.43 | 0.520 | 1.01 |
| Catalyst concentration (wt.%) * Time (h) | 1 | 97.04 | 97.04 | 7.80 | 0.013 | 1.01 |
| Error | 17 | 211.58 | 12.45 | | | |
| Total | 26 | 3896.00 | | | | |
| S = 3.52785 | $R^2$ = 94.57% | | $R^2_{(adj)}$ = 91.69% | | $R^2_{(pred)}$ = 85.88% | |

The influence and correlation of all control factors to dimethyl selectivity are presented in a Pareto chart (Figure 3), drawn based on the obtained regression polynomial equation (Equation (5)). The standardized effect achieved the minimum level for showing the influence of each variable. In this case, the size of the effect was also consistent with the length of the bar. Control factors are predicted to have a statistically significant effect and play an important role in the response if the bar of the standardized effect exceeds the minimum limit, in this case 2.11, presented in the form of the vertical red dotted line.

The VIF discloses how the assessed coefficient variance is inflated, as entailed by the multicollinearity occurring in the model. No significant multicollinearity was observed for all factors tested, as the VIF was in the interval {1, 1.03}.

Regression equation in uncoded units:

$$DS = -7.41 + 0.514\,T - 1.08\,C + 15.55\,\tau - 0.003148\,T^2 - 1.603\,C{\cdot}\tau \tag{2}$$

where:

$DS$ is dimethyl selectivity (mol%),

$T$ is temperature (°C),

$C$ is concentration (wt.%),

$\tau$ is time (h).

Figure 4a–i show the impact of individual control factors on the values of the selectivity of the transformation to dimethyl. The course of the function shows that the selectivity of dimethyl took the smallest values for the lowest temperature, shortest reaction time, and lowest concentration. An increase in all control parameters: temperature, reaction time, and concentration, resulted in an increase in dimethyl selectivity. This is a directly proportional relationship, but for temperature the optimum was observed in a range of 80–90 °C.

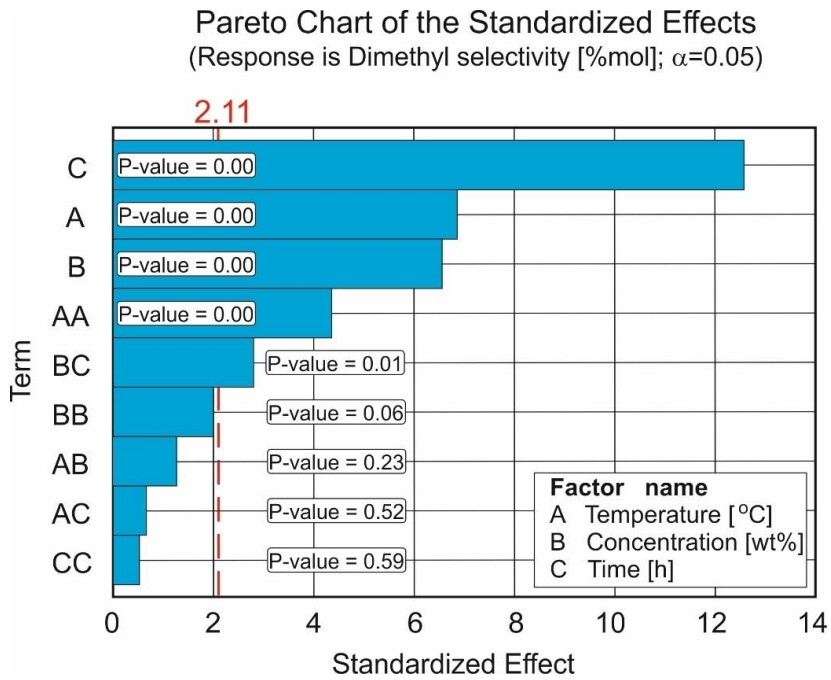

**Figure 3.** Levels of important and not important factors of the dimethyl selectivity for α = 0.05.

The maximum value of dimethyl selectivity, over 45 mol%, was reached for the maximum time, the maximum catalyst concentration, and the maximum temperature. The smallest selectivity levels were reached for the minimum catalyst concentration, lowest temperature, and lowest reaction time.

### 2.3. Impact of Control Factors on Thumbergol Selectivity

A detailed thumbergol selectivity analysis was performed by ANOVA for 95% confidence at α = 0.05 (Table 3). The factors of the model were considered significant when their *p*-values exceeded the 0.05 level. In this case, too, the coefficients $R^2$ and $R^2_{adj}$ were used to establish the accuracy of the model. The $R^2$ coefficient was 0.985 and $R^2_{adj}$ was 0.977, as shown in Table 3. The model explained 97.7% of the variance of the data, and additionally, differences between $R^2$ and $R^2_{adj}$ were equal to 0.08 for all the response variables. The response surface very accurately reflected the data.

**Table 3.** Analysis of variance of thumbergol selectivity.

| Source | DF | Adj SS | Adj MS | F-Value | *p*-Value | VIF |
|---|---|---|---|---|---|---|
| Model | 9 | 2183.85 | 242.65 | 125.72 | 0.000 | - |
| Linear | 3 | 1775.02 | 591.67 | 306.54 | 0.000 | - |
| Temperature (°C) | 1 | 13.20 | 13.20 | 6.84 | 0.018 | 1.02 |
| Catalyst concentration (wt.%) | 1 | 0.04 | 0.04 | 0.02 | 0.890 | 1.03 |
| Time (h) | 1 | 1760.84 | 1760.84 | 912.29 | 0.000 | 1.02 |
| Square | 3 | 111.14 | 37.05 | 19.19 | 0.000 | - |
| Temperature (°C) * Temperature (°C) | 1 | 1.93 | 1.93 | 1.00 | 0.332 | 1.00 |
| Catalyst concentration (wt.%) * Catalyst concentration (wt.%) | 1 | 108.96 | 108.96 | 56.45 | 0.000 | 1.02 |
| Time (h) * Time (h) | 1 | 0.25 | 0.25 | 0.13 | 0.721 | 1.01 |
| Two-Way Interaction | 3 | 165.19 | 55.06 | 28.53 | 0.000 | - |
| Temperature (°C) * Catalyst concentration (wt.%) | 1 | 28.84 | 28.84 | 14.94 | 0.001 | 1.01 |
| Temperature (°C) * Time (h) | 1 | 81.64 | 81.64 | 42.30 | 0.000 | 1.01 |
| Catalyst concentration (wt.%) * Time (h) | 1 | 54.71 | 54.71 | 28.35 | 0.000 | 1.01 |
| Error | 17 | 32.81 | 1.93 | | | |
| Total | 26 | 2216.67 | | | | |
| S = 1.3893 | $R^2$ = 98.52% | | $R^2_{(adj)}$ = 97.74% | | $R^2_{(pred)}$ = 95.77% | |

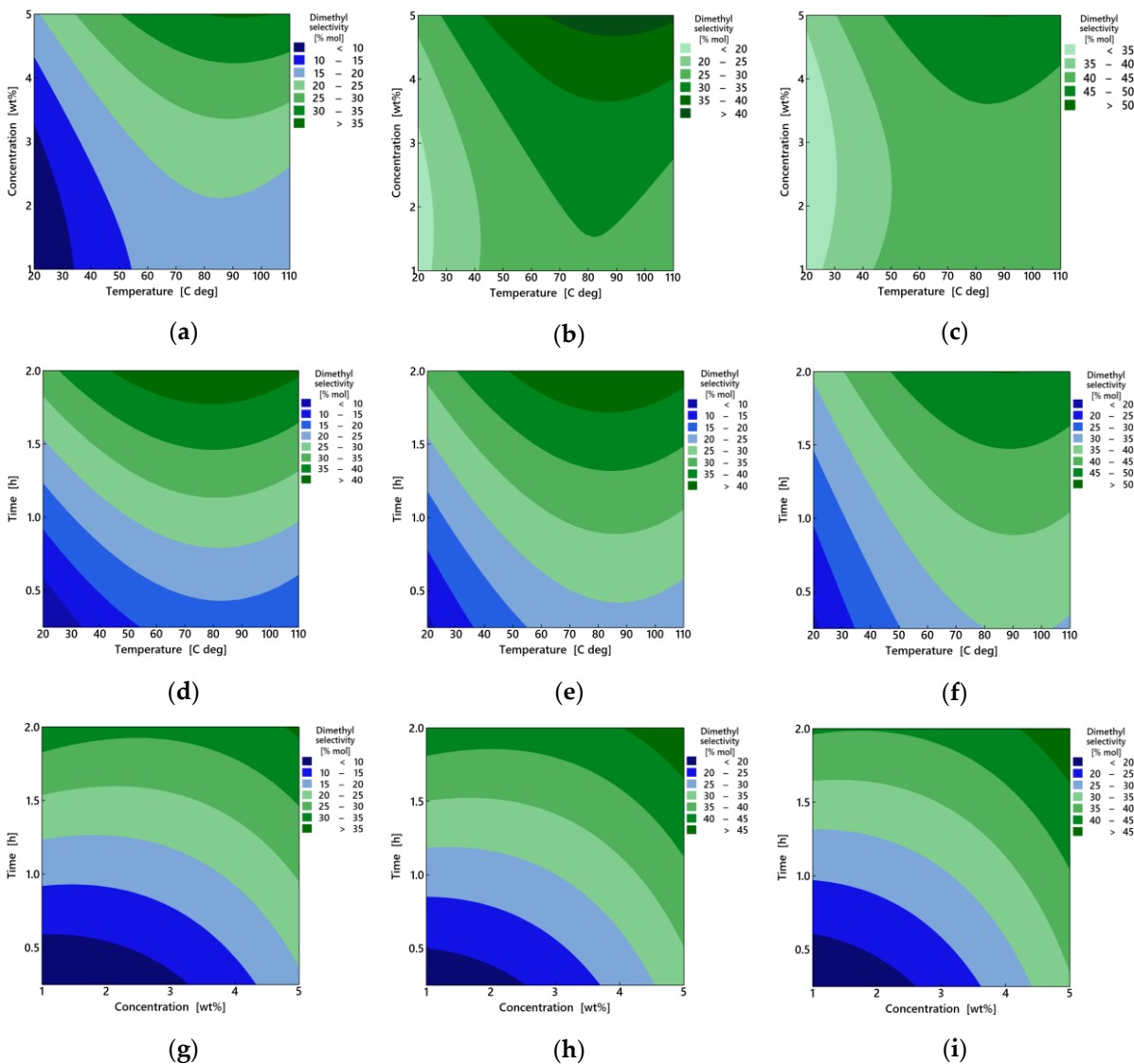

**Figure 4.** Contour plot of dimethyl selectivity at: time: (**a**) 0.25 h, (**b**) 1.125 h, and (**c**) 2 h, concentration: (**d**) 1 wt.%, (**e**) 3 wt.%, and (**f**) 5 wt.%, and temperature: (**g**) 20 °C, (**h**) 65 °C, and (**i**) 110 °C.

Based on the obtained regression polynomial equation (Equation (3)), the Pareto chart was drawn (Figure 5). It presents the influence and correlation of all control factors for thumbergol selectivity and the standardized effect achieved at the minimum level of the influence of each variable. Here, the level of the effect is shown as the length of the bar. Independent variables are predicted to have a statistically significant effect and play a significant role in the response if the bar of the standardized effect exceeds the minimum limit, in this case 2.11, presented in the form of the vertical red dotted line.

The VIF discloses how the assessed coefficient of variance is inflated, as entailed by the multicollinearity occurring in the model. No significant multicollinearity was observed for all factors tested, as the VIF was in the interval {1, 1.03}.

Regression equation in uncoded units:

$$TS = 13.75 + 0.0691\,T + 18.65\,\tau + 1.148\,C^2 - 0.01701\,T{\cdot}C - 0.0659\,T{\cdot}\tau - 1.204\,C{\cdot}\tau \quad (3)$$

where:

$TS$ is thumbergol selectivity (mol%),
$T$ is temperature (°C),
$C$ is concentration (wt.%),

$\tau$ is time (h).

Figure 6a–i demonstrate the impact of the control factors on the thumbergol selectivity level.

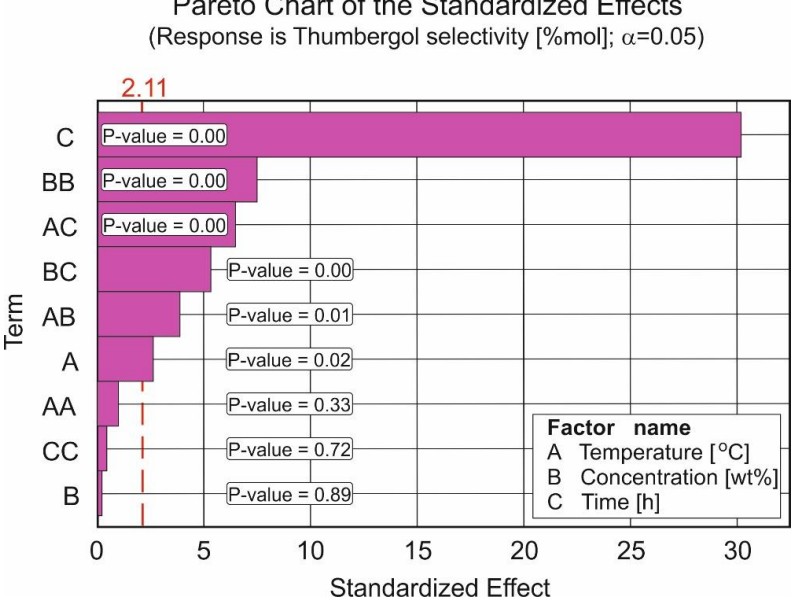

**Figure 5.** Pareto chart of the standardized effects of thumbergol selectivity for $\alpha = 0.05$.

In the case of the geraniol transformation, the influence of the concentration on thumbergol selectivity was noticeably optimum in the 3–4 wt.% range. The increase in temperature and time led to an increase in thumbergol selectivity. The maximal values of the thumbergol selectivity (over 44 mol%) were reached for the maximal temperature and for the 3.5 wt.% catalyst concentration at the maximum reaction time of 2 h. In these conditions, the minimum thumbergol selectivity level at the minimum catalyst concentration, minimum temperature, and minimum reaction period were noted.

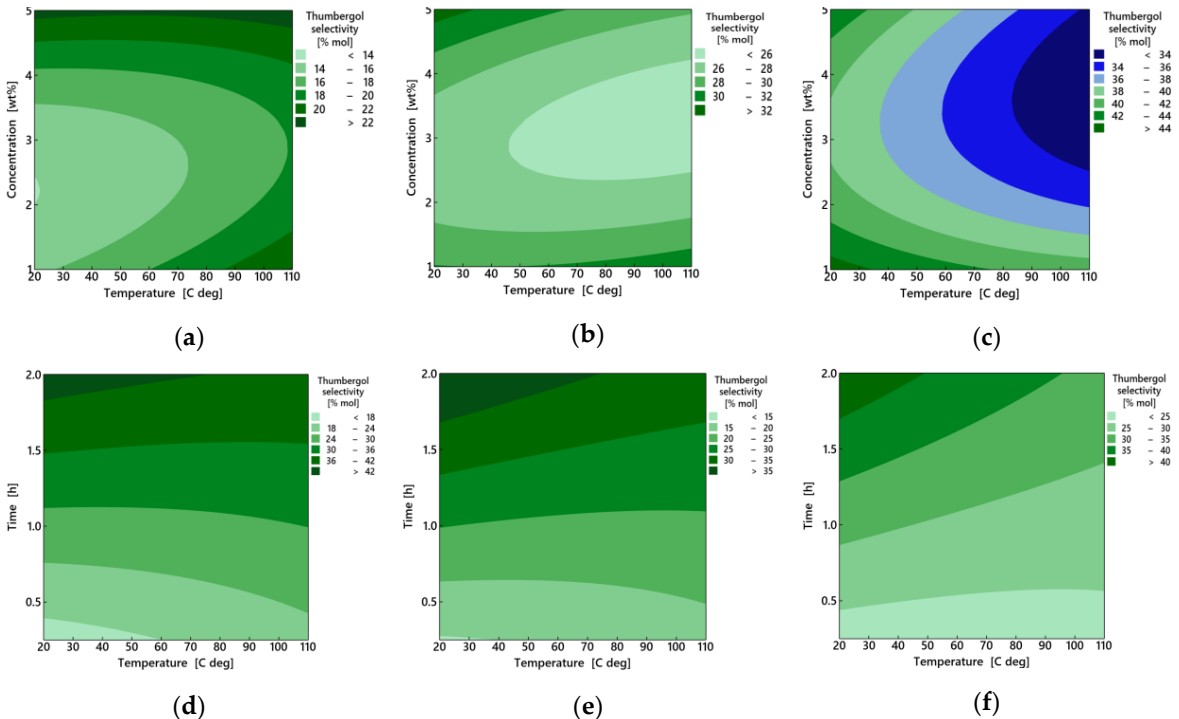

**Figure 6.** *Cont.*

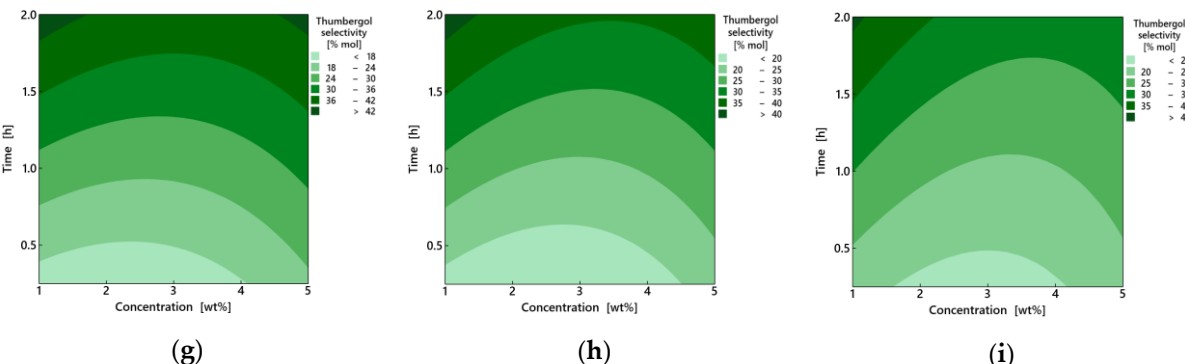

(**g**)  (**h**)  (**i**)

**Figure 6.** Contour plot of thumbergol selectivity at: time: (**a**) 0.25 h, (**b**) 1.125 h, and (**c**) 2 h, concentration: (**d**) 1 wt.%, (**e**) 3 wt.%, and (**f**) 5 wt.%, and temperature: (**g**) 20 °C, (**h**) 65 °C, and (**i**) 110 °C.

## 2.4. Composite Desirability Coefficient

The outcomes of each control factor's impact on all output factors, calculated based on Equations (3) and (4), are presented in Figure 7. Furthermore, individual and composite desirability were evaluated. Individual and composite desirability appraise how much a variable implements the designated reaction objectives.

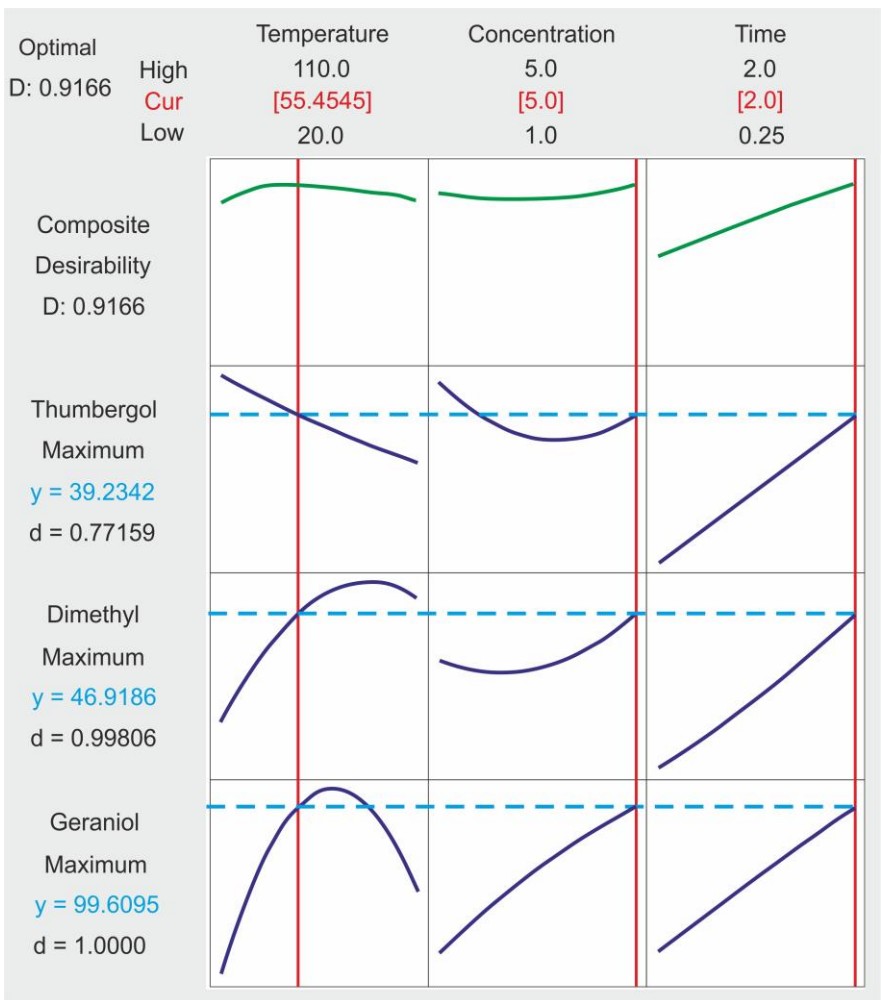

**Figure 7.** Control factors at optimal levels for each output factor.

Single desirability (d) shows how the settings can optimize an individual response. Composite desirability (D) estimates how the settings optimize a set of responses overall.

Values of desirability were taken from the interval {0, 1}, where 0 expresses the best case and 1 shows that one or more responses exceeded their acceptable restrictions.

Here, the composite desirability reached 1 (or near 1), which suggests that the process at these settings reached sensible results for each response. Based on this graph, the optimum for all output factors was defined. The best set of control factors (temperature equal to 55.5 °C, catalyst concentration equal to 5 mol%, and reaction time equal to 2 h) is shown by the red lines.

## 2.5. Discussion of the Compatibility of Experimental and Modeled Data

The high values of all the coefficients of determination ($R^2$) and only slightly different values from the raw data fit very well with the regression line. The scatter plot (Figure 8) demonstrated this compliance very well. The points are close to the straight red line. This suggests that the developed mathematical models of the geraniol conversion, dimethyl selectivity, and thumbergol selectivity can be regarded as satisfactory.

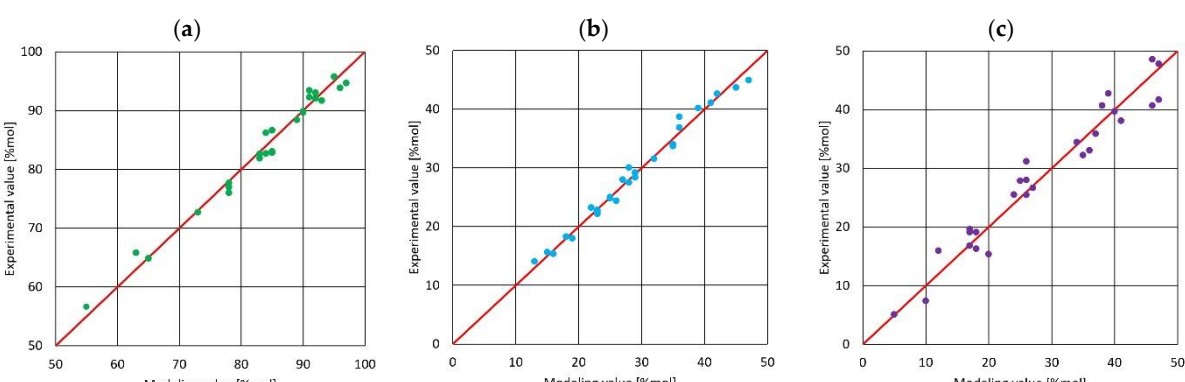

**Figure 8.** Scatter plots for each output factor: geraniol conversion (**a**), dimethyl selectivity (**b**), and thumbergol selectivity (**c**).

The optimization carried out by the RSM allowed to obtain the most favorable conditions for the geraniol isomerization process in the presence of mironecton as a catalyst with the following control parameters:

- Temperature, 55 °C,
- Catalyst concentration, 5 wt.%,
- Reaction time 2 (h).

With such parameters, the following values of selectivity of the obtained products were obtained: $C_{GE} = 99.56$, $S_{DC} = 47.77$ (mol%), and $S_{TH} = 40.47$ (mol%). The predicted optimal values from the equations were confirmed by experiments, conducted under the above-specified control parameters. The effects are shown in Figure 9. The relative difference for each response at the optimal level of the control factors at the optimal condition did not exceed 5%.

Comparing the obtained results with the results obtained from the preliminary tests [33], it can be observed that in optimal conditions the temperature value decreased from 110 to 55 °C, which was associated with lower energy expenditure.

A similar trend was observed in the case of the catalyst concentration, where there was a two-fold decrease from 10 to 5 wt.%, which had a positive effect on reducing the cost of the process.

An additional advantage is also the higher selectivity of the thumbergol to over $S_{TH} = 40$ mol% in relation to the preliminary tests (only 36 mol%), which due to its properties/applications is a more desirable product.

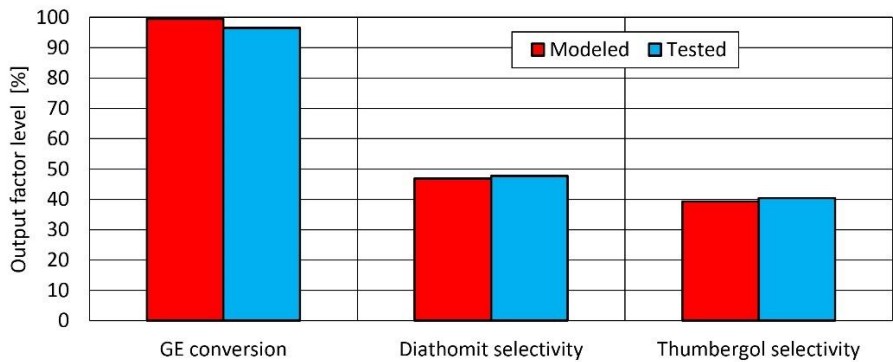

**Figure 9.** Comparison of modeled and tested output parameters of optimization.

## 3. Materials and Methods

### 3.1. Method of Transformations of Geraniol and Analyses of the Post-Reaction Mixtures

The syntheses were carried out in a glass reactor with a capacity of 25 cm$^3$, which was equipped with a reflux condenser and a magnetic stirrer with a heating function (Figure 10). The ranges of the studied parameters were as follows: temperature 80–150 °C, catalyst content 5–15 wt.%, and a reaction time from 15 min to 24 h. To perform the qualitative and quantitative analyses, the sample of the post-reaction mixture was first centrifuged and then dissolved in acetone in the ratio 1:3.

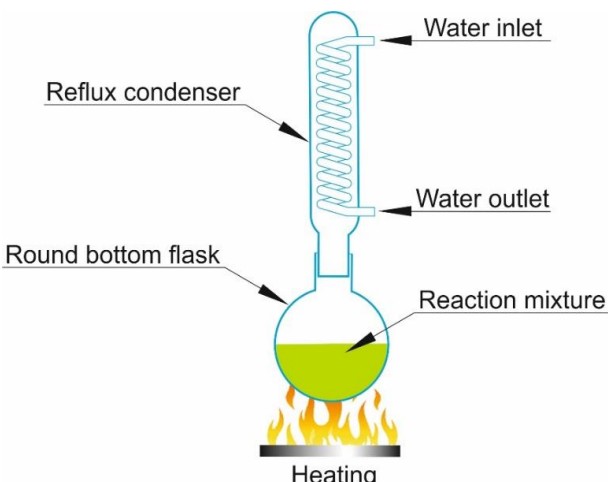

**Figure 10.** Scheme of the apparatus for carrying out the syntheses connected with the transformations of geraniol.

Qualitative analyses were performed using the GC-MS method on a ThermoQuest apparatus with a Voyager detector and a DB-5 column (filled with phenylmethylsiloxanes, 30 m × 0.25 mm × 0.5 mm). Analysis parameters were as follows: helium flow 1 mL/min, sample chamber temperature 200 °C, detector temperature 250 °C, oven temperature—isothermally for 2.5 min at 50 °C, then heating at the rate of 10 °C/min to 300 °C. Quantitative analyses were performed with the Thermo Electron FOCUS chromatograph with an FID detector and a TR-FAME column (cyanopropylphenyl packed, 30 m × 0.25 mm × 0.25 mm). The analysis parameters were as follows: helium flow 0.7 mL/min, sample chamber temperature 200 °C, detector temperature 250 °C, oven temperature—isothermally for 7 min at 60 °C, then heating at the rate of 15 °C/min to 240 °C. The FID temperature was kept at the level of 250 °C.

To define specific syntheses, the subsequent process equations were used:

1. Geraniol (GA) conversion, $C_{geraniol}$:

$$C_{geraniol} = \frac{AMGC}{AMGIR} \cdot 100\% \tag{4}$$

where:

    AMGC—amount of moles of geraniol consumed,

    AMGIR—amount of moles of geraniol introduced into the reactor.

2.     Selectivity to the key products (thumbergol—**TH** and 6,11-dimethyl-2,6,10-dodecatrien-1-ol—**DMC**), $S_{product/geraniol}$:

$$\mathbf{S}_{\frac{product}{geraniol}} = \frac{AMP}{AMGC} \cdot 100\% \tag{5}$$

where:

    AMP—amount of moles of product,

    AMGC—amount of moles of geraniol consumed.

### 3.2. Test Method

The control parameters and their ranges were as follows: temperature 20–110 °C, catalyst concentration (mironecuton) 1.0–5.0 wt.%, and reaction time 0.25–2 h. The choice of such ranges of variation was made based on previous experiments, an analysis of the state of the issue, as well as on the possibility of achieving the technological parameters of carrying out the research.

To reduce the number of tests and shorten the research period, the methodology of design of experiment (DOE) was used. The experiments were conducted according to a full factorial plan. The central composite model, included in the RSM, was used. Central composite designs can fit a full quadratic model because these designs can include information from a correctly planned factorial experiment. It consists of 27 tests (Table 4). Each of the tests conducted was repeated three times. The individual variables and the examined response of the process analysis were performed by ANOVA for 95% confidence at $\alpha = 0.05$.

**Table 4.** Details of the conducted tests.

| Test No. | Temp | Catalysts Concentration | Time | GA Conversion | DMC Selectivity | TH Selectivity |
|---|---|---|---|---|---|---|
| - | (°C) | (wt.%) | (h) | (mol%) | (mol%) | (mol%) |
| 1 | 20 | 1.0 | 0.25 | 51 | 5 | 15 |
| 2 | 20 | 1.0 | 1.00 | 63 | 18 | 27 |
| 3 | 20 | 1.0 | 2.00 | 78 | 36 | 47 |
| 4 | 20 | 2.5 | 0.25 | 65 | 10 | 13 |
| 5 | 20 | 2.5 | 1.00 | 73 | 17 | 25 |
| 6 | 20 | 2.5 | 2.00 | 85 | 26 | 39 |
| 7 | 20 | 5.0 | 0.25 | 78 | 17 | 23 |
| 8 | 20 | 5.0 | 1.00 | 83 | 24 | 32 |
| 9 | 20 | 5.0 | 2.00 | 90 | 37 | 45 |
| 10 | 60 | 1.0 | 0.25 | 78 | 12 | 19 |
| 11 | 60 | 1.0 | 1.00 | 85 | 27 | 29 |
| 12 | 60 | 1.0 | 2.00 | 92 | 39 | 42 |
| 13 | 60 | 2.5 | 0.25 | 84 | 18 | 16 |
| 14 | 60 | 2.5 | 1.00 | 89 | 26 | 26 |
| 15 | 60 | 2.5 | 2.00 | 90 | 47 | 36 |
| 16 | 60 | 5.0 | 0.25 | 91 | 35 | 23 |
| 17 | 60 | 5.0 | 1.00 | 95 | 41 | 29 |
| 18 | 60 | 5.0 | 2.00 | 97 | 47 | 36 |
| 19 | 110 | 1.0 | 0.25 | 83 | 20 | 23 |
| 20 | 110 | 1.0 | 1.00 | 84 | 26 | 28 |
| 21 | 110 | 1.0 | 2.00 | 93 | 38 | 41 |
| 22 | 110 | 2.5 | 0.25 | 85 | 17 | 18 |
| 23 | 110 | 2.5 | 1.00 | 90 | 25 | 25 |
| 24 | 110 | 2.5 | 2.00 | 91 | 46 | 35 |
| 25 | 110 | 5.0 | 0.25 | 92 | 34 | 22 |
| 26 | 110 | 5.0 | 1.00 | 96 | 40 | 28 |
| 27 | 110 | 5.0 | 2.00 | 97 | 46 | 35 |

RSM is a statistical and mathematical combination method of modeling [34,35]. Statistica software was used to create the model equations. The impacts of the control factors' (independent variables) conversion into geraniol as well as dimethyl and thumbergol selectivity (dependent variables) are shown in Table 4.

Columns 2–4 present the values of the control parameters (inputs) for the research process. The following columns (5–7) show the values of the results (output parameters).

The second-degree equation for determining the regression model value is:

$$y = \beta_0 + \sum_{i=1}^{k} \beta_i x_i + \sum_{i=1}^{k} \beta_{ii} x_i^2 \pm \varepsilon \tag{6}$$

where:

$y$ is the dependent variable (response),

$x_i$ shows values of the *i*-th cutting parameter,

$\beta_0$, $\beta_i$, and $\beta_{ii}$ are the factors of regressions,

$\varepsilon$ is the error acquired in the cutting.

## 4. Conclusions

The use of the RSM in the geraniol transformation process made it possible to determine which of the process parameters studied significantly affected the reaction, while permitting the disregard of those factors that had only a marginal impact on the effects. The optimization of these control factors allowed to establish their values for obtaining the maximum values of the studied functions, and to find the interactions among the factors of the studied functions. The studies performed here on optimization allowed to obtain the following conclusions:

- For geraniol conversion, the optimal value was obtained at over 99 mol% at a temperature equal to 71.8 °C, a catalyst concentration equal to 5 mol%, and a reaction time equal to 2 h.
- For dimethyl selectivity, the optimal value was reached at 45 mol% at a temperature equal to 20 °C, a catalyst concentration equal to 1 mol%, and a reaction time of almost 2 h.
- For thumbergol selectivity, the optimal value was obtained at 50.1 mol% for a temperature equal to 87.3 °C, a catalyst concentration equal to 5 mol%, and a reaction time equal to 2 h.
- The optimal set of control factors for all output factors characterized were a temperature equal to 55.5 °C, a catalyst concentration equal to 5 mol%, and a reaction time equal to 2 h.

The use of RSM allowed to facilitate the research procedures, decreasing the time to procure relevant results, as well as lowering the cost of research by reducing the necessary number of tests.

**Author Contributions:** Conceptualization, A.F.-B.; methodology, A.F.-B.; software, A.P. and A.R.-Z.; validation, A.P. and A.R.-Z.; formal analysis, A.F.-B.; investigation, A.F.-B.; resources, A.F.-B.; data curation, A.R.-Z.; writing—original draft preparation, A.F.-B., A.P. and A.R.-Z.; writing—review and editing, A.F.-B. and A.R.-Z.; visualization, A.P.; supervision, A.R.-Z.; project administration, A.R.-Z.; funding acquisition, A.F.-B. and A.R.-Z. All authors have read and agreed to the published version of the manuscript.

**Funding:** This research received no external funding.

**Data Availability Statement:** Not applicable.

**Conflicts of Interest:** The authors declare no conflict of interest.

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
