# Peer review of "Modeling and Optimization of Geraniol ((2E)-3,7-Dimethyl-2,6-Octadiene-l-ol) Transformation Process Using Response Surface Methodology (RSM)"

_catalysts, doi:10.3390/catal13020320_

Round 1

Reviewer 1 Report (Previous Reviewer 2)

The quality of revised manuscript has been improved significantly according to the comments. Now I’m satisfied with the revision made by the authors. Therefore, this manuscript is recommended for publication in the present form. 

Author Response

Thank you for your positive comments.

Reviewer 2 Report (New Reviewer)

I have a serious question regarding the thesis, so I request revision.

1. The above paper is a research paper, and although there may be a brief description or explanation of the theory, the explanation of RSM in the introduction compared to the total amount is too unnecessary, including many descriptions.

Please completely restructure the predicate.

(The composition of the written research is in the form of textbooks for learning RSM, not research papers.)

2. It was explained that three experimental conditions (Temp, Catalysis concentration, Time) were used and that the central composition design was used.

Why is the number of tests 27?

Your study setup is a three-level full factorial design.

Three-level full factorial design is DOE but not RSM

3. In your research result, there are 3 dependent variables

In almost all results, the maximum, minimum, and inflection points do not fall within the range you set.

Explain the rationale for setting the range for an independent variable.

Round 2

Reviewer 2 Report (New Reviewer)

The results of experimental design through methods and principles that are too different from the standards known to reviewers are presented.

The above paper is an experimental design using GENERAL FULL FACTORIAL included in DOE.

The above result can check the point for each point, but cannot predict other points. For prediction, an RSM with all measured distances between centers, poles, and factors is required. The design is different from the above result, and the corresponding range cannot be measured.

  The FULL FACTORIAL DESIGN was analyzed by the RSM method, and the 3D FIGURE was derived by looking at the ANOVA results.

It is necessary to check whether the method can be used and whether there are papers or previous studies that calculated in this way.

Author Response

This manuscript is a resubmission of an earlier submission. The following is a list of the peer review reports and author responses from that submission.

Round 1

Reviewer 1 Report

1. The abstract needs to be improved. Also, It is recommended to include the intervals of each factor at least.

2. Some acronyms are not defined (e.g., VIKOR, AFIS, ANN...)

3. Introduction is mainly focused on RSM. However, this methodology is well known and depth details about it are not needed in the introduction. On the contrary, Introduction should focus on the chemical process, which is the main problem of the study. Therefore, the information on the critical variables of geraniol production is scarce and is only justified by a previous study (ref 36).

4. The difference between this study and the work cited in reference 34 must be made clear.

5. Experimental setup used to obtain the experimental data must be detailed.

6. Examples of the use of RSM in previous studies are well; however, there are too many of them, turning Introduction section into a boring text.

7. Some superscripts and subscripts are missing (e.g., line 144, 167…).

8. The use of the term “s” in equations 8 and 9 is unnecessary, these equations should be reviewed.

9. Authors must specify why they used a central composite model and not another. In addition, it must be specified if repetitions were made.

10. In table 2, experiments are ordered. Authors should explain the method of randomization, which is mandatory for RSM.

11. Methodology is so general and important details are missing. Some relevant questions cannot be answered with the information presented in the methodology: how experimental data was obtained? What optimization software was used? What statistical variables were used to fit the models?

12. The manuscrit is disorganized.

13. Figure 1 does not show "p-value". Pareto chart should be discussed properly. Please review DOI: 10.1016/j.heliyon.2022.e11546

14. Table 1 is not necessary (this information could be included in the text or in another table).   

15. 3D plots are better at showing optimal spots than 2D (contours) plots. Please review DOI: 10.3390/en14248366

16. what could be discussed about difference between R2, R2(adj) and R2(pred)?

17. Paper just describes results of RSM; but, a comparison of these results with others reports in literature is not made, nor is the phenomenology of the process analyzed. Then, RSM is a powerful mathematical and statistical tool; but, the results of RSM are also required to be analyzed from a scientific and processes point of view.

Reviewer 2 Report

1.     The full definition is not provided in the first use of EDM, VIKOR, AFIS, ANN, UAN, GA, RMSE, SEP and AAD.

2.     The RSM is widely applied. Please also cite the application of RSM in electronic packaging in line #99.

3.     Please re-highlight the main focus and brief method at the end of the introduction.

4.     What criteria are used to consider the control factors' value range? Please briefly explain in the manuscript.

5.     Why is the level 2 value not a midpoint between levels 1 and 3? For example, 65 is a midpoint value between 20 and 110, 3 for a value between 1 and 5, and 1.13 for values between 0.25 and 2.0. Please justify in the manuscript.

6.     The software used to generate the DOE combination is not mentioned.

7.     What is the alpha value used in the CCD method? Please include it in the methodology.

8.     The replication is not included in table 2. How many replications are considered?

9.     In Tables 4 and 5, please explain why using the square model for equations 11, 12  and 13 since the Catalyst concentr and time P-value is more than 0.05. Why not use linear and 2-way interaction models to describe the response equation?

10.  The most significant factors for three response are not mentioned.

11.  The percentage of the discrepancy between the experimental data (in Table 2) and the predicted value from equations 11-13 is not analyzed.

12.  How do authors ensure the predicted equations and the optimal value obtained is reliable without any final confirmation of the experimental data?

13.  What is the percentage of error for the response if using the setting of the value of the optimal factor in the experiment?